# Influence of Welding Speed on Microstructure and Mechanical Properties of 5251 Aluminum Alloy Joints Fabricated by Self-Reacting Friction Stir Welding

**DOI:** 10.3390/ma14206178

**Published:** 2021-10-18

**Authors:** Shikang Gao, Li Zhou, Guangda Sun, Huihui Zhao, Xiaolong Chu, Gaohui Li, Hongyun Zhao

**Affiliations:** 1State Key Laboratory of Advanced Welding and Joining, Harbin Institute of Technology, Harbin 150001, China; 17863110658@163.com (S.G.); quantasun@163.com (G.S.); 13258667016@163.com (G.L.); hy_zhao66@163.com (H.Z.); 2Shandong Provincial Key Laboratory of Special Welding Technology, Harbin Institute of Technology at Weihai, Weihai 264209, China; hitwh170820317@163.com; 3Shanghai Aerospace Equipments Manufacturer Co., Ltd., Shanghai 200245, China; vx15665325902@163.com

**Keywords:** self-reacting friction stir welding, aluminum alloy, microstructure, mechanical properties industry

## Abstract

In the present study, 8 mm-thick 5251 aluminum alloy was self-reacting friction stir welded (SRFSW) employing an optimized friction stir tool to analyze the effect of welding speed from 150 to 450 mm/min on the microstructure and mechanical properties at a constant rotation speed of 400 rpm. The results indicated that high-quality surface finish and defect-free joints were successfully obtained under suitable process parameters. The microhardness distribution profiles on the transverse section of joint exhibited a typical “W” pattern. The lowest hardness values located at the heat-affected zone (HAZ) and the width of the softened region decreased with increasing welding speed. The tensile strength significantly decreased due to the void defect, which showed mixed fracture characteristics induced by the decreasing welding speed. The average tensile strength and elongation achieved by the SRFSW process were 242.61 MPa and 8.3% with optimal welding conditions, and the fracture surface exhibited a typical toughness fracture mode.

## 1. Introduction

To the best of our knowledge, aluminum alloys have been widely applied in welding structures related to commercial products as a result of their low density, high mechanical strength, high thermal conductivity, and excellent corrosion resistance [1,2]. Welding defects of porosity, hot cracking, joint softening and alloying element loss occurred in the process of joining aluminum alloys by conventional fusion welding which deteriorate dramatically the mechanical property [3,4,5]. The defects of fusion welding can be controlled with self-piercing riveting and clinching. However, the joint produced by self-piercing riveting and clinching has poor corrosion resistance and may not be aesthetically acceptable [6,7,8]. Friction stir welding (FSW) invented at TWI in the UK in 1991, as a relatively new solid-state joining process, has become a promising option for shipbuilding, aerospace, transport and so on [9]. The benefits of FSW have been reviewed to include compatibility with all types of aluminum alloys, without concern for filler metal and shielding gas, elimination of solidification defects, arc, smoke and splash related to the fusion welding [10,11,12]. The requirement for groove, complex treatment prior to welding and residual stress and deformation of post-weld structure involved in the conventional welding method could also be limited. However, the presence of significant plunging force and rigid backing plate/anvil to react process loads makes it difficult for some complex-shaped structures to implement the friction stir welding process [13,14,15,16,17,18].

Self-reacting friction stir welding (SRFSW), as a new type of friction stir welding technology, which overcomes the shortcomings of conventional FSW, has been successfully applied to weld materials with the thickness of 3–30 mm. The microhardness values of the joint are slightly lower, and the tensile strength is comparable to that of the conventional FSWed joint owing to high heat input [19]. The SRFSW process shares similarities with conventional FSW but with distinct differences that could be attributed to the tool configuration [20]. The tool of SRFSW is characterized by a pin connected with two shoulders, ensuring the upper and lower surfaces of workpieces are in contact with the two shoulders, respectively. The presence of a bottom shoulder as opposed to the backing plate lowers or eliminates the axial force and saves the cost of manufacturing the rigid clamping [21]. In addition, the flexibility of assembling and welding can be increased, which allows for three-dimensional welding of complex hollow-shaped structures. During the welding process, both shoulders rotate with the pin and interact with the materials to be welded. The upper and lower surfaces of the workpieces are welded simultaneously in elimination of the incomplete root penetration defects fundamentally.

At present, there have been plenty of efforts to investigate on the SRFSWed aluminum alloy. Some literature was focusing on the design of the novel tool, and the additional torque and three-dimensional stress engaged on the tool allows for the tool with higher strength and wear resistance. As reported by Chen et al. [22], a sound self-reacting friction stir welded joint was obtained under the appropriate conditions with the combined use of convex and concave tools. Ahmed et al. [23] reported the influence of different pin geometries on temperature distribution and self-reacting friction stir welded joints’ properties. The results show that the square pin produces a higher temperature and tensile shear properties than the other pins. A method of securing two different types of refractory material to provide a composite FSW tool was described in Ref. [24], and 25 mm-thick aluminum alloy material and 8 mm-thick 12% chromium steel were SRFSWed by means of this tool. However, the employment of the newly designed tool produced a welded joint with more refined microstructures and higher mechanical properties, and the investigations on the microstructures and mechanical properties contributed to understanding the various characteristics of the joint and providing a theoretical basis for the development and optimization of welding process parameters. Liu et al. [25] pointed out the grain size of the nugget zone (NZ) increased with the increasing welding speed and the band patterns from the AS to the weld center were also observed in the NZ. Furthermore, plenty of β’ phases, which decreased as the welding speed increased appeared again in the HAZ. The effect of the welding parameters on microstructures and mechanical properties of ASRFSW-ed 6082-T6 aluminum alloy was researched by Threadgill and Sued et al. [18,26]. Celik et al. [27] reported the effect of self-reacting friction stir welding at various welding rates and rotational speeds. The tensile strength of welding joint can reach 95.17% of the base metal at the optimal parameters.

As we can see from previous literatures, the studies on the influence of the welding parameters on the microstructures and mechanical properties were primarily focused on the 2xxx and 6xxx series [28]. However, few works in the literature dealing with 5xxx series were reported and further investigation is necessary. The 5xxx aluminum alloy, being non-heat treatable, has been widely utilized in manufacturing the automobile body due to its high corrosion resistance, ductility and low density. In this paper, the 5251 aluminum alloy was welded employing the SRFSW process with a fixed-gap tool, and the influence of welding speeds on the microstructure and the mechanical properties was investigated in detail.

## 2. Experimental Procedure

The 5251 aluminum alloy rolled plates with a H temper were adopted with the dimensions 300 × 100 × 8 mm, and the nominal chemical composition and mechanical properties of 5251 aluminum alloy are presented in Table 1. 

An optimization designed SRFSW tool made of SKD61 tool steel was characterized by two unequal concave shoulders with upper and lower diameters of 26 mm and 24 mm, respectively, and a cylindrical, tetra-flat pin with diameter of 12 mm schematically shown in Figure 1.

Prior to welding, the surface oxide film and contamination of the base material (BM) was removed by mechanical polishing, and then the surface was wiped with acetone. As shown in Table 2, an FSW machine in position control was applied to fabricate the butting joint with a constant rotation speed of 400 rpm and various welding speed ranging from 150 mm/min to 450 mm/min. All the joints were cross-sectioned perpendicularly to the welding direction for metallographic analysis by using an electrical-discharge cutting machine, followed by the standard metallographic procedure. The polished samples were etched by Keller’s reagent for 1–2 min, and the grains morphology and the precipitates distributions were observed using an Olympus-DSX500 optical microscopy (OM) and a TESCAN scanning electron microscopy (SEM).

The Vickers hardness measurement points, as presented in Figure 2, were divided into three rows along the plate thickness at 1 mm intervals in the long transverse direction across the metallographic sections on a HMAS-D1000Z hardness tester using a load of 100 g for 10 s. Transverse tensile test specimens were cut perpendicular to the welding direction from the joints with the dimension prepared according to China National Standard GB/T228-2002 [30]. In order to improve the results reliability, three samples were prepared for tensile testing under each welding parameter. The tensile test was carried out on the Instron-1186 mechanical tester at a displacement speed of 3 mm/min. The tensile fracture features were analyzed by an SEM equipped with energy dispersive X-ray spectroscopy (EDS).

## 3. Results and Discussion

### 3.1. The Surface Appearance of the Welded Joint

The surface appearance of the weld was observed by visual method, and the macroscopic characteristics and defects were noticed. It is worth mentioning that the side where the tool rotation direction is the same as the tool travel direction (opposite the direction of metal flow) is the advancing side (AS), and the side where the tool rotation direction is opposite the tool travel direction (parallel to the direction of metal flow) is the retreating side (RS).

As shown in Figure 3, the surface appearance morphology of the joint was obtained at various welding speeds from 150 mm/min to 450 mm/min. A series of experiments were made by changing the welding speed, and it can be noted that the welding speed which is lower than 150 mm/min tends to cause serious flash and the welding speed which is more than 450 mm/min tends to cause the stirring pin to fracture. Regarding all the joints, the weld was well-formed, and the upper and lower surfaces were smooth and continuous without grooves defects observed. The viscosity of plasticized metal with high temperature was lowered due to the excessive heat input at the welding speed of 150 mm/min, leading the formation of the “extrusion die” which was surrounded by the upper and lower shoulders, the pin and the BM. This obviously caused the plastic metal to flow out from the weld and the formation of mushroom-shaped flash on the RS. The heat input was reduced in elimination of the flash at the highest welding speed of 450 mm/min.

Furthermore, the distance between the arc lines increased with the increasing welding speed, which was consistent with Equation (1) [31].
(1)$$d=k_{d}*\frac{v}{\omega *f}$$

The eccentric squeeze induced by the shoulders is of vital importance for the arc lines formation in the SRFSWed aluminum alloy. It can be deduced from Equation (1) that the distance between the arc lines is relevant to the factor of arc lines space (*k_d_*), the welding speed (*v*), the rotation speed (*ω*) and friction coefficient (*f*) between the shoulders and the BM. The arc lines space increases as the welding speed increased and decreases with the increasing rotation speed and friction coefficient.

### 3.2. The Cross-Sectioned Appearance of the Welded Joint

As depicted in the Figure 4, cross-section morphology at varied welding speed reveals that there existed large cavity defect at a low welding speed of 150 mm/min and the cavity defect area decreased with the increasing welding speed. This could be explained by the fact that the plastic flow of metal increases caused by the presence of Si in the 5251 aluminum alloy and the thermal exposure increases owing to the low welding speed, resulting in the cavitation effect and the formation of the void defect [32]. With the welding speed increasing to 450 mm/min, the thermal exposure is decreased and the sufficient plastic flow occurs in the “extrusion die”. The defect-free joint could be obtained at a welding speed of 450 mm/min. However, the heat input is too low to produce the continuous weld owing to the insufficient flow of the plasticized metal as the welding speed is more than 450 mm/min.

Figure 4c reveals the typical cross-section morphology at the welding speed of 350 mm/min which presents the four distinctive zones: nugget zone (NZ), heat-affected zone (HAZ), thermo-mechanically affected zone (TMAZ) and base metal (BM). The NZ could be divided into NZ-top, NZ-bottom and NZ-middle [33]. It can be seen that the geometry of the NZ displays a “dumbbell-shape” with a width of the surface equivalent to the shoulder diameter and the width at the mid thickness more than the diameter of the pin. Such a shape was obtained as the material in the NZ was directly subjected to the intense friction of the stirring pin and affected by thermal conduction from the upper and bottom shoulders, which increases the material flow and allows the plastic metal to be fully stirred within a certain distance away from the pin. The thermal conduction from the upper and bottom surfaces to the mid-thickness of the cross-section simultaneously plays a positive effect on the more uniform heat distribution, as well as improving the microstructure and performance of the joint. It can be deduced from the schematic diagram of the plastic material flow (See Figure 5) that this exhibits two flow fields in opposite directions at different distances from the tool on the AS, and the flow direction of the plastic material on the RS are consistent, causing a blurred boundary between TMAZ on the RS and the NZ, and the area transits more smoothly. Furthermore, a band pattern was located in the middle NZ from the AS to the NZ center, which can be attributed to the movement of plasticized materials adhering to the pin from the RS to the AS and the convergence of the upper and lower flow fields with different speeds.

### 3.3. Microstructure in the Typical Areas of the Joint

A typical cold rolled structure was exhibited with elongated and coarse grains in BM, as displayed in Figure 6a. The grains in the HAZ which are similar to that in BM are only affected by the thermal fields, rather than experiencing the stirring and extrusion effect of the tool, thus no deformation and dynamic recrystallization (DRX) occurred, as shown in Figure 6b. The TMAZ is the transition zone between the NZ and HAZ, which experienced the thermal cycle and the extrusion effect of the pin. However, the insufficient stirring effect resulted in the formation of bent and elongated grains rather than the broken grains. In addition, the boundary between the TMAZ on the AS and the NZ was much clearer in contrast to the RS, and the deformation of the grains was more obviously oriented (See Figure 6c,d).

The NZ is the area where the grains experience frictional heat and the stirring effect, and the microstructure varies significantly compared with the BM, as shown in Figure 6e–g. The refined grains of the NZ were randomly oriented, and the plastic metal was broken into refined DRXed equiaxed grains at a high deformation rate induced by the thermal cycle of peak temperature and the mechanical stirring of the tool. Since the upper and lower shoulders generated heat simultaneously, the heat conduction was sufficient along the thickness direction, and the grain size in the three areas was similar. As presented in Figure 6h, the grains of the band pattern are marked in a white dashed line in Figure 4c were elongated with a certain flow direction, which was slightly coarser than the equiaxed grains in other parts of the NZ. On the one hand, this can be explained by remembering that the convergence of the upper and lower flow fields with different speeds reduced the frictional heat from the shoulders. On the other hand, the sheared and accumulated metal were released to the band pattern due to the utilization of the tetra-flat pin profile, which improved the ability of transferring the material and weakened the stirring effect. Hence, the deformation and temperature required by the DRX were insufficient, and the elongated grains were obtained under the forging action of the shoulders. Moreover, the lazy “S” line was observed in the NZ (See Figure 4d) and Figure 6i exhibited the micrograph of “S” line at high magnification. The reason for the lazy “S” line is that the oxide film on the plates surface was not cleaned completely and the re-oxidation occurred during welding. Previous researches indicated that the “S” line had a bad effect on the joint fatigue performance and the total length of “S” line is related with the tensile strength. The lazy “S” line in a nearly linear geometry had a negative influence on the yield strength of the SRFSWed joint, and the strength of the weld decreased to a quarter of that of BM [28].

For a further observation, SEM analysis was used to examine the grain structures and the particle distribution features. As shown in Figure 7a–d, the grain orientation in the typical areas of the joint under SEM is consistent with that observed under OM. Large amounts of dark particles were observed in Figure 6, and SEM images in Figure 7 indicate that they could be corrosion pit owing to the corrosion dissolution of precipitated particles in the etching process. To identify the precipitated particles, the sample was only polished and the white particles were observed, as depicted in Figure 7e. The compositions of these precipitates were obtained and the ratio of Mg/Si was about 2:1, thus reflecting the existence of Mg_2_Si.

### 3.4. Effect of the Welding Speed on the Microstructure

Microstructural evolution is in large part influenced by the welding speed, and the typical microstructure at varied welding speed is depicted in Figure 8. The linear intercept method was adopted for the quantitative measurement of the equiaxed grains sizes in NZ. According to the measurement of the grain size, the average grain size in the NZ is 13.5 μm approximately at the welding speed of 150 mm/min. As the welding speed increases to 350 mm/min, the average grain size decreases to 12.5 μm. The grain size is more refined about 10 μm at the welding speed of 450 mm/min. It is well known that the grain size is determined by the heat input and strain rate, and both are lowed with increasing welding speed during FSW. On the one hand, the decreasing strain rate accounts for the coarser re-crystallized grains. On the other hand, the decreasing heat input plays a positive effect on the grain refinement. It can be concluded from the NZ grain size that the heat input plays a decisive role in the finer re-crystallized grains. Microstructure in TMAZ on the AS of the joint is shown in Figure 8d–f. We can conclude that the TMAZ was characterized by the refined and elongated grains as the welding speed increased from 150 mm/min to 450 mm/min. Results indicates that the decreasing thermal input caused by increasing welding speed is beneficial to grain refinement in the TMAZ in a certain range. Furthermore, the microstructure in TMAZ on the RS had the same trend, thus the optical microstructures were not repeated. With respect to the microstructure in HAZ on the AS, the increasing welding heat brought on the lower welding speed would lead to the coarser microstructure (See Figure 8g–i). While the welding speed was gradually increased to 450 mm/min, the heat exposure was continuously reduced and the grain coarsening was weakened, that is to say, the grain size was reduced in some degree. The typical microstructure in HAZ on the RS was consistent with that on the AS.

### 3.5. Microhardness Distribution across the Joint

The SRFSWed joint is a heterogeneous composite, and the microhardness distribution reflects the evolution of microstructure and mechanical properties in varied areas of the welded joint. Consistent with results previously discussed, the microhardness profiles exhibit W-shape. The hardness in BM displays the highest value, and the minima positions in these profiles correspond to the HAZ, where the AS and the RS are slightly different. The hardness value of the NZ is higher than that of the HAZ and TMAZ, but are still significantly lower than that of the BM. Except for the increased location destiny, the improvement of microhardness has to do with the presence of the refined equiaxed grains according to the Hall–Petch relation (2).
(2)$$\Delta \sigma_{H-P}=\frac{K_{y}}{\sqrt{D}}$$
where *K_y_* is the constant, and *D* is the average diameter of the grains. Moreover, the solid solution strengthening occurred in the NZ associated with the dissolution of Mg_2_Si particles, which contribute to high microhardness in this region [34]. Comprehensively, the hardness in the NZ is increased. The highest microhardness is located in the BM, which can be attributed to the strain hardening due to the cold-rolling process [35]. The coarse grains of the HAZ were not stirred by the pin, and the precipitated particles aggregated and grew under the thermal cycle. That can be explained by the fact that the microhardness in the HAZ is significantly reduced. Partial DRX and recovery occurred in the TMAZ, accompanying the formation of bent, strengthened grains. Therefore, the hardness values in the TMAZ lie between the NZ and HAZ.

The 2D hardness map of the joints in the cross section at different welding speed are shown in Figure 9. It could be concluded that the lowest hardness is located in the HAZ. As the welding speed increased from 150 mm/min to 450 mm/min, the thermal exposure was reduced and the softened zone of the joint was gradually narrowed, which resulted in the lowest hardness value improving from 52 HV to 58 HV. This phenomenon could be explained by the shrink of the thermal field and the rapid thermal cycle of the soften metal with the increase of welding speed, decreasing the extent of over-aging.

### 3.6. Tensile Properties of the Joint

As displayed in Figure 10, the tensile test results were obtained at the rotation speed of 400 rpm and varied welding speeds. It can be seen that the tensile strength of the joint gradually increased as the welding speed increased from 150 mm/min to 450 mm/min, and the maximum tensile strength of 242.61 ± 8.33 MPa was obtained at the welding speed of 450 mm/min with the joint efficiency equal to 76%, which is attributed to the disappearance of the void defects, the grain refinement and the softened area reduction in width. The variation tendency of the joint elongation with welding speed is similar to that of the tensile strength, and the maximum elongation was 8.3% at the welding speed of 450 mm/min. It could also be noted from Figure 10 that the stirring pin was fractured as the welding speed further increased to 550 mm/min owing to the fact that the heat input was too low to result in sufficient plastic flow, thus the pin suffered from high shear strength while the welds remained too cold.

As mentioned earlier, the cavity defects were obtained in the joints as the process parameters were unreasonable, and the defect-free joint was obtained at the welding speed of 450 mm/min. Figure 11 shows the engineering stress–strain curve of the joint and the corresponding fracture location as the welding speeds were 350 (sample 1–3) and 450 mm/min (sample 4–6), respectively. It can be seen from sample 1–3 that the joints were fractured at the cavity defect under this parameter, and the strengthening stage of elastic deformation and plastic deformation also existed. However, no obvious “necking” phenomenon was observed before the joint was fractured. In addition, the maximum values of the three curves have large difference, indicating that the stability of the welding quality is poor. Due to the lower microhardness values, the defect-free joint failed in the HAZ on the RS, and the curves of sample 4–6 show that the joint first elastically deformed, and the stress increased linearly with strain. Having entered the yield strengthening stage, obvious plastics deformation occurred and the tensile force increased to a maximum of 48 kN with the corresponding tensile stress of 242.6 MPa. Subsequently, the joint entered the local plastics deformation stage, that is, “necking”, until it completely failed. The curves of the three samples under this parameter are similar, indicating the high stability of the weld quality.

Figure 12 illustrates the SEM micrographs on the fracture surfaces using the rotation speed of 400 rpm and welding speed of 350 mm/min. The delamination phenomenon was observed at the fraction surfaces. At the beginning of the tensile test, the greater stress concentration around the cavity defect promoted the fracture initiation and propagated along the weaken zone. At this time, the plastic deformation of the material was insufficient, and plenty of small glide planes and the smooth pores from which the second phase divorced could be observed on the fracture surface (see Figure 12b), which indicated a typical brittle fracture mode; The massive dimples and tear ridges were obtained after the crack crossed over the transition region, exhibiting a ductile fracture mode. The analysis shows the fracture surface exhibited toughness-brittleness mixed fracture characteristics under this parameter.

For the defect-free joint, as depicted in Figure 12d–f, the joint was uniformly fractured and has no delamination phenomenon with plenty of dimples distributed in the fracture surface. Furthermore, the second phase particles existed in the dimples, and several point ID EDS studies were conducted to determine the composition of the second phase particles with a characteristic one being shown in Figure 12f. It can be deduced from the detected elements and the Mg/Si ratio that the white particles could be Mg_2_Si particles. The fracture mechanism could be explained as dislocation pile-up occurred around Mg_2_Si particles, which resulted in the microcrack. The microcrack aggregated, propagated and finally the joint was fractured, presenting a typical ductile fracture mode.

## 4. Conclusions

In this study, the 8 mm 5251 aluminum alloy was SRFSWed using an optimized tool, and the influence of the welding speed on the surface appearance, microstructure and mechanical properties was analyzed. Based on the above results, the following conclusions could be drawn:Under all the experimental parameters, the upper and lower surfaces were smooth without micro-defects observed. The appearance of flash could be avoided with the increasing welding speed. The joint with maximal strength efficiency was obtained at the welding speed of 450 mm/min and rotation speed of 400 rpm.The dumbbell-shaped cross-section of the joint was composed of four varied areas: the NZ, the TMAZ, the HAZ and the BM. The microstructure in the NZ was refined DRXed equiaxed grains, and the grains were further refined as the welding speed increased. Dynamic recovery occurred in the deformed microstructure of the TMAZ, the size of grains decreasing with the increasing welding speed. The coarser grains were elongated in the HAZ with the decrease in the grains size as the welding speed increased.The softening effect occurred in the SRFSWed joint, and the microhardness distribution exhibited a typical “W” pattern. The softened area of the joint was narrowed and the lowest microhardness value increased by increasing the welding speed. Tensile test indicated that the tensile property was worse when induced by the cavity defect with the fracture surfaces exhibiting mixed fracture characteristics, as the welding speed was lower. With the welding speed increasing to 450 mm/min, the corresponding tensile strength and elongation were reached to the maximum values of 242.6 MPa and 8.3%, respectively, and the fracture analysis presented a typical ductile fracture mode.

## Figures and Tables

**Figure 1 materials-14-06178-f001:**
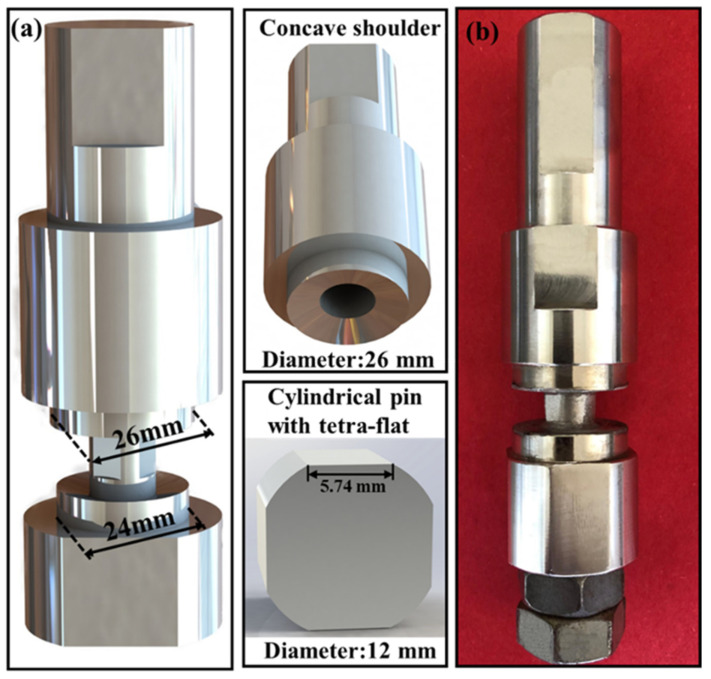
Details of the self-reacting friction stir welding tool: (**a**) characteristic sizes of the tool system; (**b**) actual image of the tool.

**Figure 2 materials-14-06178-f002:**
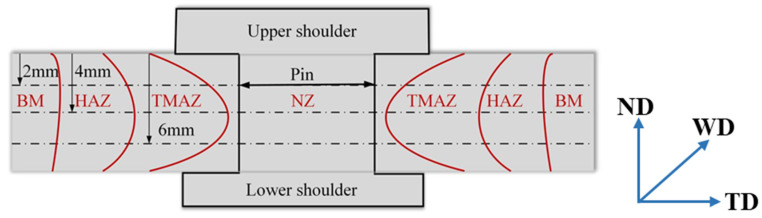
Schematic illustration for the microhardness measurement points.

**Figure 3 materials-14-06178-f003:**
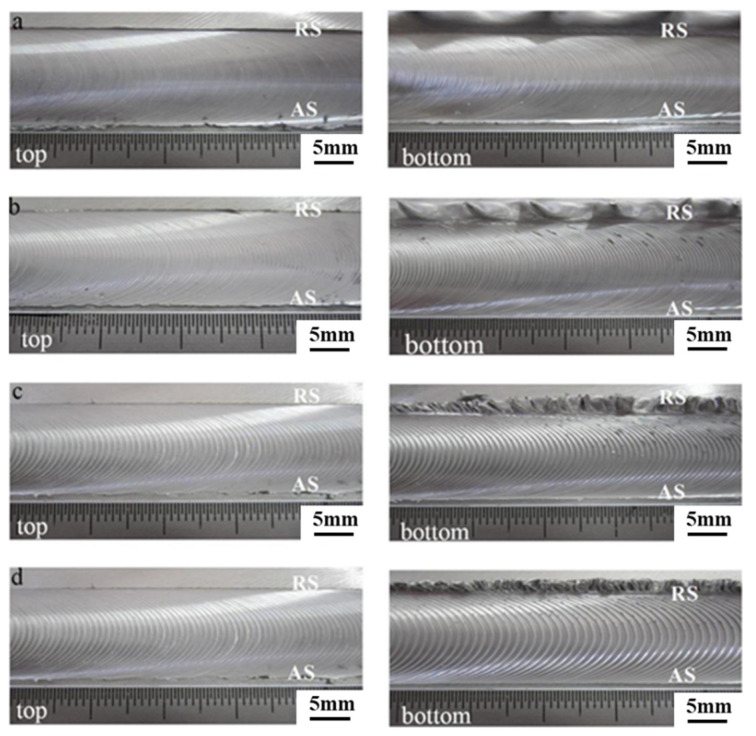
Surface appearance morphology of the joint welded at the welding speed of: (**a**) 150 mm/min; (**b**) 250 mm/min; (**c**) 350 mm/min and (**d**) 450 mm/min.

**Figure 4 materials-14-06178-f004:**
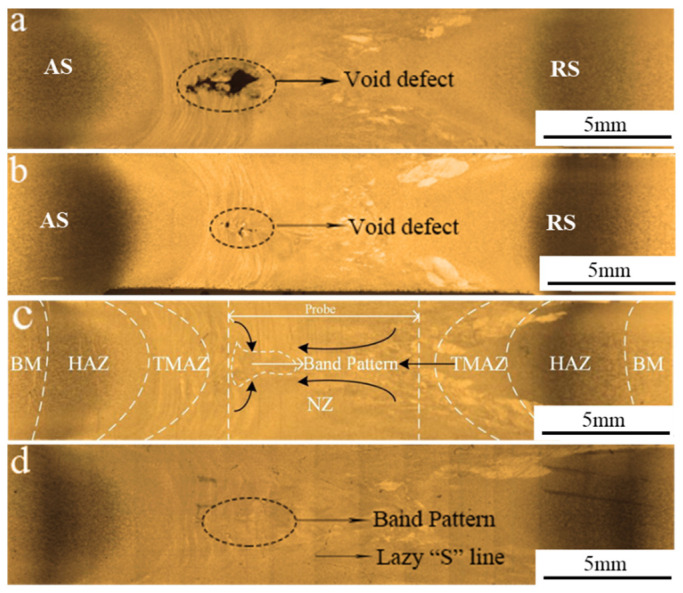
Cross-section morphology at welding speeds of: (**a**) 150 mm/min; (**b**) 250 mm/min; (**c**) 350 mm/min and (**d**) 450 mm/min.

**Figure 5 materials-14-06178-f005:**
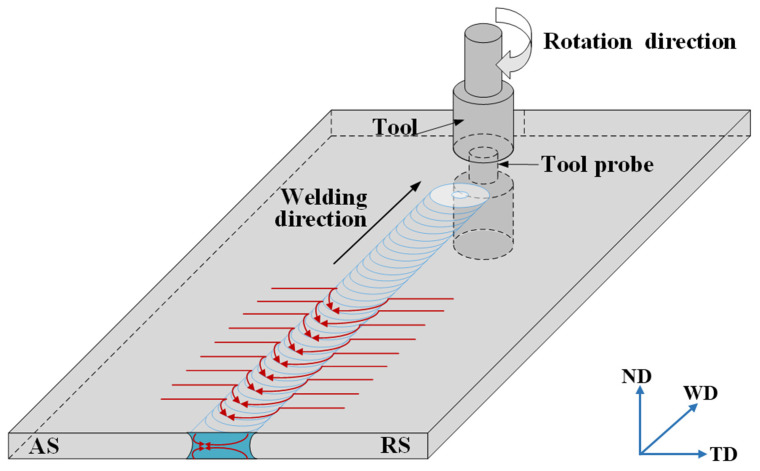
The schematic diagram of the plastic material flow in the horizontal direction.

**Figure 6 materials-14-06178-f006:**
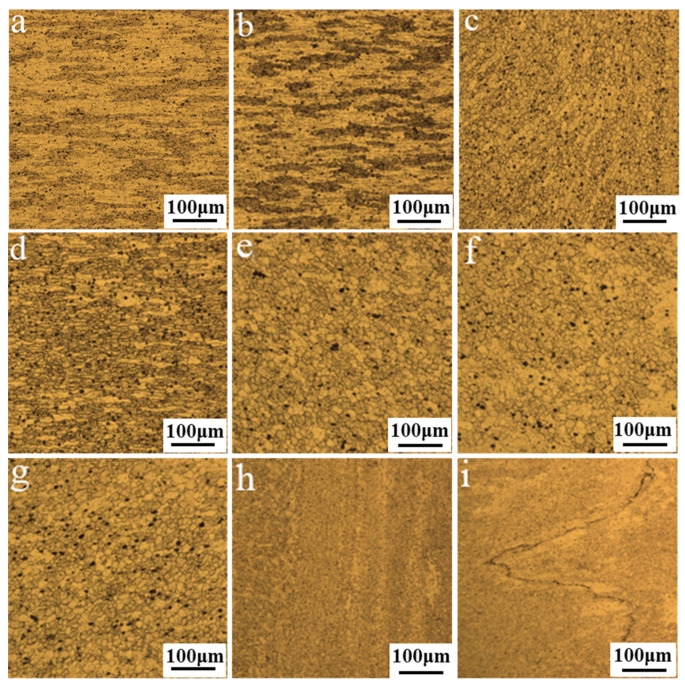
Microstructure under OM at the rotation speed of 400 rpm and welding speed of 350 mm/min in the section of: (**a**) base material (BM); (**b**) heat-affected zone (HAZ); (**c**) thermo-mechanically affected zone (TMAZ) on the AS; (**d**) TMAZ on the retreating side (RS); (**e**) the upper nugget zone (NZ); (**f**) the central NZ; (**g**) the lower NZ; (**h**) the band pattern; and (**i**) the “S” line.

**Figure 7 materials-14-06178-f007:**
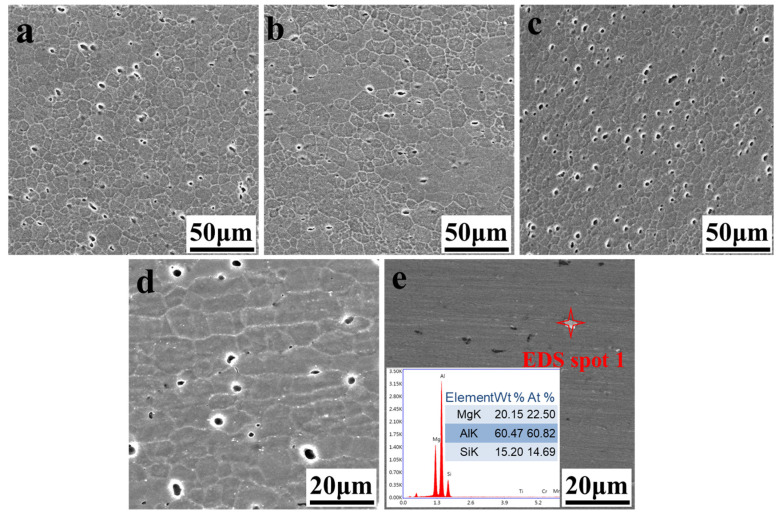
Microstructure under scanning electron microscope (SEM) at the rotation speed of 400 rpm and welding speed of 350 mm/min in the section of: (**a**) NZ; (**b**) TMAZ on the AS; (**c**) TMAZ on the RS; (**d**) HAZ and (**e**) energy-dispersive spectroscopy (EDS) result of the precipitated particles.

**Figure 8 materials-14-06178-f008:**
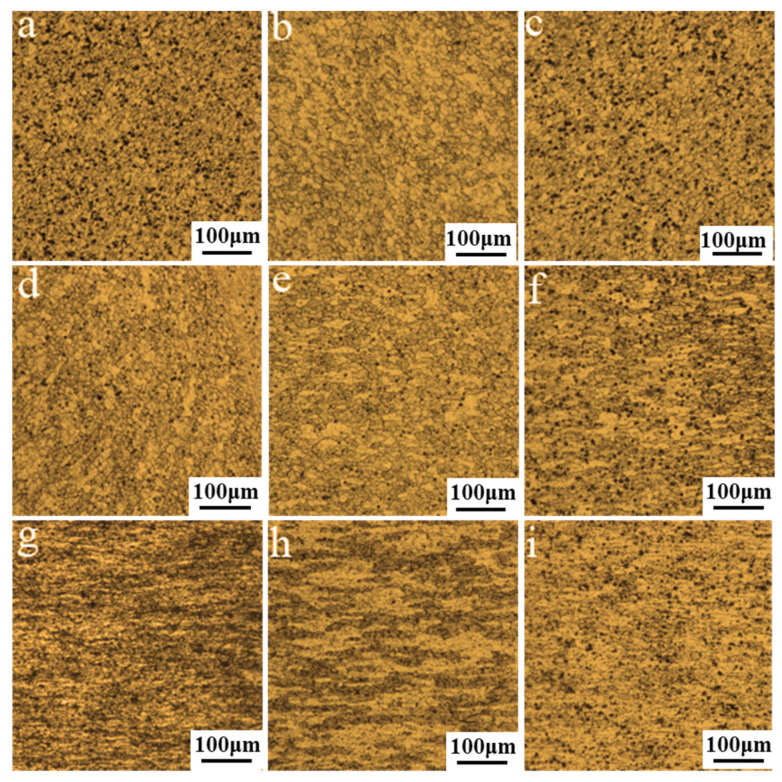
The microstructure of the joint in the section of: (**a**) NZ in the 150 mm/min joint; (**b**) NZ in the 250 mm/min joint; (**c**) NZ in the 450 mm/min joint; (**d**) TMAZ in the 150 mm/min joint; (**e**) TMAZ in the 250 mm/min joint; (**f**) TMAZ in the 450 mm/min joint; (**g**) HAZ in the 150 mm/min joint; (**h**) HAZ in the 250 mm/min joint and (**i**) HAZ in the 450 mm/min joint.

**Figure 9 materials-14-06178-f009:**
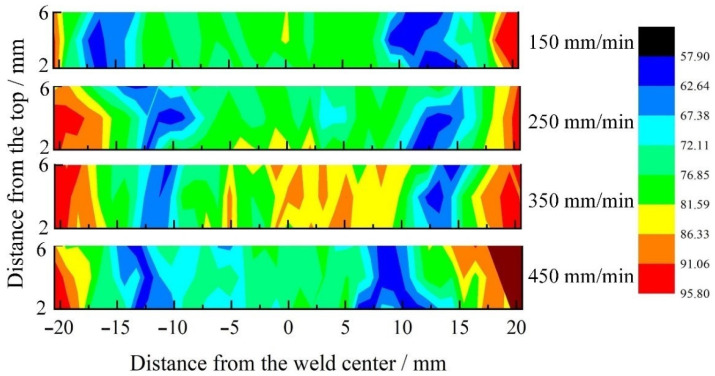
2D hardness maps through the joints at a rotation speed of 400 rpm and different welding speed.

**Figure 10 materials-14-06178-f010:**
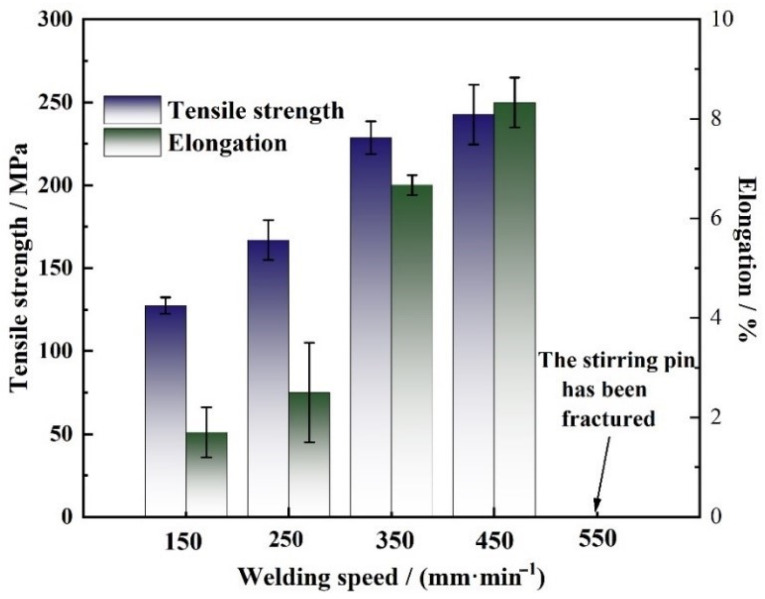
Tensile properties of the joints welded at different welding speeds.

**Figure 11 materials-14-06178-f011:**
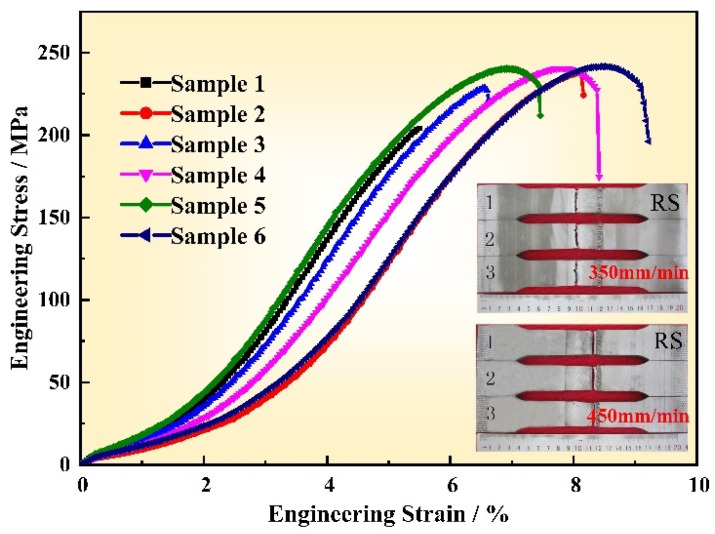
The tensile stress-strain curve of the joint at the rotation speed of 400 rpm and welding speed of 350 mm/min (sample 1–3) and 450 mm/min (sample 4–6).

**Figure 12 materials-14-06178-f012:**
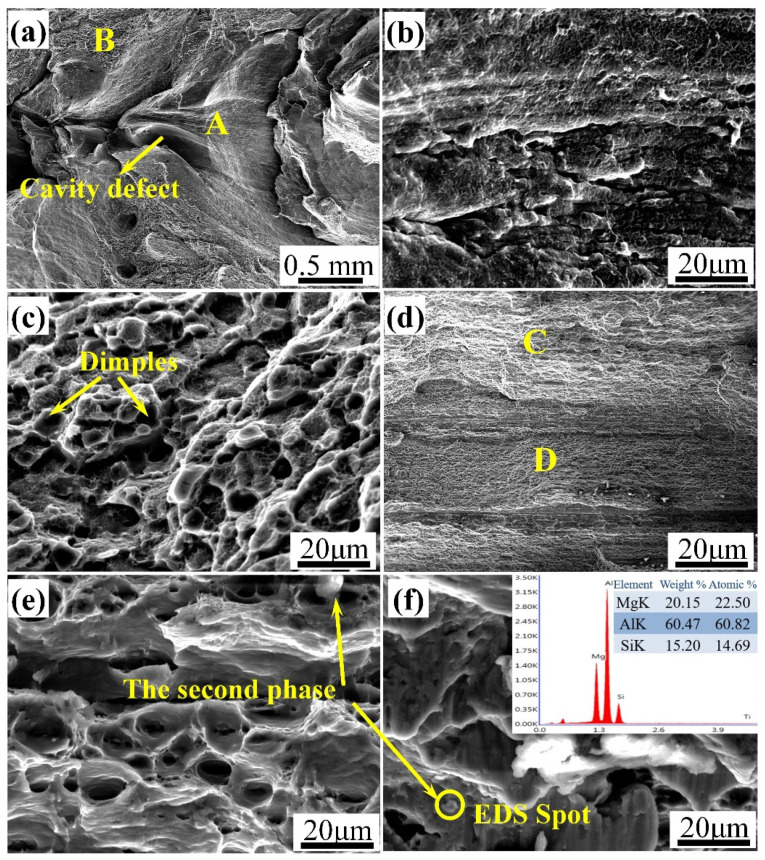
SEM micrographs showing fracture surfaces of the joints: (**a**) the macro-morphology at low magnification at the welding speed of 350 mm/min; (**b**) high magnification images of region A; (**c**) high magnification images of region B; (**d**) the macro-morphology at low magnification at the welding speed of 450 mm/min; (**e**) high magnification images of region C; (**f**) high magnification images of region D and EDS of the precipitated particles.

**Table 1 materials-14-06178-t001:** Chemical Composition of 5251 Aluminum Alloy [29].

Chemical Composition/wt%								
Al	Ti	Mg	Si	Fe	Zn	Cu	Mn	Cr
Bal.	0.01	1.9	0.15	0.4	0.3	0.1	0.5	0.02

**Table 2 materials-14-06178-t002:** The Experimental Conditions.

Reference	Welding Speed mm/min	Rotation Speed rpm	Penetration Depth of Shoulder mm
[1]	150	400	0
[2]	250	400	0
[3]	350	400	0
[4]	450	400	0

## Data Availability

Not applicable.

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
