# Peer review of "Influence of Welding Speed on Microstructure and Mechanical Properties of 5251 Aluminum Alloy Joints Fabricated by Self-Reacting Friction Stir Welding"

_materials, 2021, doi:10.3390/ma14206178_

Round 1
Reviewer 1 Report
The study “Influence of welding speed on microstructure and mechanical properties of 5251 aluminum alloy joints fabricated by self-reacting friction stir welding” was reviewed and the following decision was made. The current research work corresponds well to the journal scope. The results are novel and of scientific value. The study can be published after proper correction.
Comments.
(1) “Prior to welding, the surface oxide film and contamination of the base material (BM) should be removed, and the surface was wiped with acetone” The oxide layer on the Al surface is formed fast and cannot be removed with acetone.
(2) “It could be explained that plastic flow of metal increases caused by the presence of Si in the 5251 aluminum alloy and the thermal exposure increases owing to the low welding speed” How can authors explain the increase of plastic flow by presence of Si?
(3) “…the maximum tensile strength of 242.6 MPa was obtained at…” the error bars should be added for the strength values. How many samples authors used for each state? It seems doubtful that the accuracy of the measurement is in second decimal places MPa.
Reviewer 2 Report
This paper presented a laboratory experiment on the joining of aluminum 5251 alloys by self-re-acting friction stir welding. The joint quality was examined by hardness profile measurement, microstructure, and tensile strength. It was concluded that defect-free joints were produced by the favorable joining parameters and a typical “W” pattern was observed for the hardness profile. The joint tensile strength was 242 MPa as opposed to the tensile strength of the base metal of 300 MPa. The results were drawn based on experimental observation and the paper will attract readers in this subject area. However, the following needs to be addressed before publication:
1. The reviewer is not convinced with the literature review. Only 4 references were cited for the recent years between 2018-2021. A couple of hundred paper was found in Scopus search on this topic. More recent articles should be cited. Also, to increase the readership in the broad area of dissimilar metal joining, please include other joining techniques such as self-piercing riveting and clinching and cite the following two references:
Quality of self-piercing riveting (SPR) joints from cross-sectional perspective: A review- Archives of Civil and Mechanical Engineering-Volume 18
Research on the Influence of the AW 5754 Aluminum Alloy State Condition and Sheet Arrangements with AW 6082 Aluminum Alloy on the Forming Process and Strength of the ClinchRivet Joints, Materials 2021, 14(11), 2980
2. Table 1 is presented before it was present in the text.
3. Figure 1: The details are missing. Please provide all the dimensions. It is impossible to reproduce the experiment with the given dimensions.
4. Please include the directions (transverse and long) in figure 2.
5. The strain rate seems to be too high 3mm/min for the tensile test. Please include the reason for choosing this specific strain rate. Did you follow any specific reference/standard?
6. A scale is added in figure 3. However, it is not clear which scale was used. Is it an mm scale or an inch scale?
7. The experimental conditions are not clearly mentioned in the experimental section. Please provide a table showing all the experimental conditions used for this experiment.
8. Why only one microstructure is provided for the speed of 350 mm/min? Do you expect the same microstructure for other speeds?
The paper is organized poorly. It needs a massive restructure to make it more readable.
Round 2
Reviewer 2 Report
The authors addressed all the issues raised by the reviewer. It is ready to be published.